# The Bright Side of Skin Autofluorescence Determination in Children and Adolescents with Newly Diagnosed Type 1 Diabetes Mellitus: A Potential Predictor of Remission?

**DOI:** 10.3390/ijerph191911950

**Published:** 2022-09-21

**Authors:** Kristina Podolakova, Lubomir Barak, Emilia Jancova, Juraj Stanik, Katarina Sebekova, Ludmila Podracka

**Affiliations:** 1Department of Pediatrics, Medical Faculty of Comenius University and National Institute for Children’s Diseases, Limbova 1, 83340 Bratislava, Slovakia; 2Institute of Experimental Endocrinology, Biomedical Research Center, Slovak Academy of Sciences, Dubravska Cesta 9, 84505 Bratislava, Slovakia; 3Institute of Molecular Biomedicine, Medical Faculty of Comenius University, Sasinkova 4, 81108 Bratislava, Slovakia

**Keywords:** skin autofluorescence, HbA1c, Type 1 diabetes mellitus, children, remission period

## Abstract

Skin autofluorescence (SAF) is a noninvasive method reflecting tissue accumulation of advanced glycation end products (AGEs). We investigated whether, in newly diagnosed children and adolescents with type 1 diabetes (T1D), this surrogate marker of long-term glycemia is associated with markers of the early manifestation phase, residual secretion capacity of the ß-cells, and the occurrence of remission. SAF was measured in 114 children and adolescents (age: 8.0 ± 4.5 years, 44% girls) at the time of T1D diagnosis, and related to HbA1c, C-peptide, diabetic ketoacidosis, and remission. 56 patients were followed up for 1 year. Seventy-four sex- and age-matched healthy individuals served as controls. SAF was higher in the T1D group compared with controls (1.0 ± 0.2 vs. 0.9 ± 0.2, *p* < 0.001). At the time of diagnosis, SAF correlated with HbA1c (r = 0.285, *p* = 0.002), was similar in patients with and without ketoacidosis, and was lower in the remitters compared with non-remitters (0.95 ± 0.18 vs. 1.04 ± 0.26, *p* = 0.027). Unlike HbA1c, SAF was an independent predictor of remission (∆R^2^ = 0.051, *p* = 0.004). Former studies consider SAF in diabetic patients as a tool to identify individuals at an increased risk of chronic complications. Here we show that determination of SAF at the time of T1D diagnosis might potentially predict remission, at least in children.

## 1. Introduction

Type 1 diabetes mellitus (T1D) is a metabolic disorder characterized by chronic hyperglycemia resulting from defects in insulin secretion, action, or both, which leads to abnormalities of carbohydrate, fat, and protein metabolism. An increase in incidence of T1D has been observed globally in recent decades [1]. Childhood and adolescence are periods during which intensive education and treatment may prevent or delay the onset and progression of T1D-associated chronic vascular complications. Clinically evident diabetes-related vascular complications in this age are rare. However, early functional and structural abnormalities may be present a few years after the onset of T1D [2].

Advanced glycation end products (AGEs) are heterogeneous compounds produced by non-enzymatic reactions between reducing sugars or reactive dicarbonyls with amino groups of proteins. Glycation significantly alters protein molecular conformation and function, such as enhancing cross-linking, altering enzyme activity, or recognition of the substrates by receptors [3,4]. AGEs accumulation is accelerated under conditions of oxidative stress, resulting in higher AGEs levels in patients with diabetes mellitus, renal or liver failure, anorexia nervosa, or in smokers [5,6,7,8,9]. 

In patients with diabetes, glycated hemoglobin (HbA1c)—an early glycation product—has been adopted as a diagnostic criterion for diabetes, and is a gold standard for glucose monitoring. The HbA1c represents the glycemic control over the past 1–3 months [10,11]. This relatively short-term monitoring has driven the search for other markers encompassing long-term glycemic control [12]. As collagen has a half-life of about 10–15 years, it is an appropriate candidate molecule to monitor a long-term accumulation of AGEs and imposed oxidative stress [13,14,15]. Because of the characteristic fluorescence of AGEs, the collagen-associated AGE-fluorescence can be non-invasively monitored on the skin [16]. In adults, skin autofluorescence (SAF) correlates with the amounts of chemically-defined AGEs in skin biopsy specimens; and in patients with T1D, SAF correlates with microvascular and macrovascular complications. Thus measurement of SAF is used as a predictor of progression of these complications [17,18,19].

Data on SAF in children and adolescents are scarce and are limited to patients exposed to diabetes for 2-to-10 years. The studies consistently report that patients present with higher SAF values compared with their age-matched healthy siblings; that SAF rises with aging and diabetes duration; that it correlates with the current and mean HbA1c over the previous period; and it associates with microvascular complications in adolescent patients [12,14,20,21].

To our knowledge, data on SAF in newly diagnosed children and adolescents with T1D are missing. To this point, we investigated SAF in individuals with newly diagnosed T1D, studied the relationships with variables characterizing the early manifestation phase of diabetes and with the residual secretion capacity of the ß-cells, and occurrence of remission. 

We hypothesized that already at the time of T1D diagnosis, SAF is higher than in healthy age-matched controls and that SAF correlates with other established markers of the further course of diabetes, including the occurrence of remission. 

## 2. Materials and Methods

### 2.1. Sample Size Calculation

Age-dependent reference ranges of SAF in healthy Slovaks were used to estimate the sample size [22]. To achieve a 10% difference in SAF between healthy controls and DM1 patients (corresponding to about 4.5-year difference in chronological age), at α = 0.05, and power of 80%, 70 subjects per group were required with the enrollment ratio set to 1:1, or 52 controls and 104 patients with DM1 considering the ratio 1:2.

### 2.2. Participants

From July 2017 to December 2020, 275 children/adolescents were newly diagnosed with diabetes mellitus at the Children Diabetes Centre at the Department of Pediatrics, Faculty of Medicine of Comenius University, and The National Institute of Children’s Diseases in Bratislava. Diagnosis of the T1D was made based on the current criteria of the International Society for Pediatric and Adolescent Diabetes (ISPAD) [1]. Other types of diabetes mellitus (neonatal diabetes with confirmed monogenic etiology, Maturity Onset Diabetes of the Young, Type 2 diabetes, and syndromic forms of diabetes mellitus) (n = 28) were excluded in the first step. There was no child with neonatal diabetes or any syndrome form of diabetes during the reporting period. For the current analysis, patients with any acute or other chronic illnesses, current smokers or those reporting household members as smokers, and members of ethnic minorities (such as Koreans, Vietnamese, etc.) were excluded in the next step. All children were finally, 114 (56% boys) white Caucasians of Central European descent aged 1-to-18 years were included into this prospective study (Figure 1). Seventy-four age- and sex-matched healthy siblings or non-siblings of the patients (51% boys) served as controls. 

A written informed consent to participate was obtained from the parents; children and adolescents gave verbal assent. The study was approved by the Ethics Committee of the National Institute of Children’s Diseases in Bratislava, Slovakia, and conducted in adherence to the tenets of the Helsinki declaration. 

### 2.3. Demographic and Clinical Data

Demographic data were collected, and SAF was measured in all participants. At the time of diabetes diagnosis, age, sex, anthropometric data, glycemia, HbA1c, fasting C-peptide levels, complete acid-based balance, and autoantibodies were recorded. Insulin dose at discharge; anthropometric data, glycemia, HbA1c, C-peptide levels, and insulin dose were recorded at the follow-up visits—every 3 months up to 1 year after the T1D diagnosis. Standard deviation score (SDS) for body mass index BMI was calculated using the sex- and age-specific Slovak reference values [23].

### 2.4. Biochemical Analyses

Laboratory parameters were measured in blood samples in the Department of Laboratory Medicine of the National Institute of Children’s Diseases in Bratislava. 

Glucose, pH, and bicarbonate levels were determined using standard laboratory methods. HbA1c was evaluated from whole blood using an HPLC method (D-10 analyzer, Bio-Rad, Hercules, CA, USA). Fasting serum C-peptide levels were determined using the electrochemiluminescence immunoassay method (Cobas e411, Roche Diagnostic, Basel, Switzerland). Pancreatic autoantibodies (against glutamic acid decarboxylase, tyrosine phosphatase, and insulin were detected using commercial ELISA kits (RSR Ltd., Cardiff, UK), according to the manufacturer’s instructions. Mean HbA1c was calculated using all obtained values during the first year of the follow-up. Delta HbA1c was calculated as a difference in HbA1c values at the time of T1D diagnosis and after one year of T1D duration. HbA1c values were transformed to the International Federation of Clinical Chemistry values (http://www.ngsp.org/ifccngsp.asp, accessed on 7 June 2008) and Diabetes Control and Complications Trial values. 

### 2.5. Skin Autofluorescence

SAF was measured on the volar side of the dominant forearm using the AGE Reader (Diagnoptics BV, Groningen, The Netherlands) (Figure 2). The autofluorescence reader illuminates a skin surface of approximately 4 cm^2^ guarded against surrounding light with an ultraviolet A light with a peak excitation of 370 nm. Emission light (λem: 420–600 nm) and reflected excitation light (with a wavelength of 300–420 nm) from the skin is measured. SAF is calculated as the ratio between the emission light and reflected excitation light, multiplied by 100 and expressed in arbitrary units (AU) [12,24]. Three independent measurements were performed for each patient with approximately 30-s intervals, and the average of three values was used as the final result. Measurements in the patients were performed on the same day as HbA1c and C-peptide at the time of T1D diagnosis. At one year, SAF was measured only in 56 patients with T1D, because only patients who had a follow-up visit exactly 12 months ± 2 weeks were included to this group. Some patients did not come in on the expected follow up date due to acute infection or due to lockdown during the COVID and visit was only with using telemedicine.

### 2.6. Hospitalization and Initiation of the Insulin Treatment

Insulin treatment was started as soon as possible after diagnosis. If diabetic ketoacidosis was confirmed (hyperglycemia > 11 mmol/L (>200 mg/dL), metabolic acidosis (pH < 7.3 and/or HCO_3_^−^ < 15 mmol/L), and ketonemia or ketonuria insulin was administered IV [25]. All children and adolescents were set on intensive insulin regimens delivered by combinations of multiple daily injections during hospitalization and therapy was individualized for each patient to achieve optimal metabolic control. Remission phase was defined according to ISPAD guidelines 2018 as HbA1c < 7% (53 mmol/mol) and the daily insulin requirement < 0.5 IU/kg/day [26]. 

### 2.7. Statistical Analysis

Variables were checked for normality using the Shapiro–Wilk test. Normally distributed data are expressed as the mean ± SD. Non-normally distributed data (C-peptide serum levels and C-peptide/glucose ratio) are expressed as the median and interquartile range. Non-normally distributed data were logarithmically transformed before further analyses. Differences between the two groups were tested using the two-sided Student’s *t*-test for normally distributed and Mann–Whitney U test for non-normally distributed metric data, and by Fisher’s test for binary data. SAF values at diagnosis and follow-up were compared using a paired two-sided Student’s *t*-test. A two-way ANOVA was used to estimate the relationship between SAF (dependent variable) and sex, the presence of T1D (factors). Univariate associations of SAF with selected variables were calculated in Pearson’s correlation and linear regression. Multivariate associations between selected variables were determined in forward linear and logistic multiple regression analyses. HbA1c change in 1 year of T1D duration, C-peptide after one year of T1D duration, and C-peptide/glucose ratio after 1 year of T1D duration were used as dependent variables for the linear regression, and remission onset as the dependent variable for the logistic regression analyses. Age at T1D diagnosis, HbA1c, and SAF at T1D diagnosis were used as co-variates for both linear and logistic regression analyses. A receiver operating characteristic (ROC) analysis was performed, with remission as a state variable and SAF as a test variable. *p* < 0.05 was considered statistically significant. Statistical analyses were performed using the SPSSv27 (IBM, Armonk, NY, USA) and GraphPad Prism 7 (GraphPad, San Diego, CA, USA), and ClinCalc.com (accessed on 6 October 2021) (ClinCalc LLC, Arlington Heights, IL, USA) software.

## 3. Results

### 3.1. Group Characteristics

Baseline characteristics of children and adolescents with newly diagnosed T1D are in Table 1. For further analyses, we divided participants into remission and non-remission groups based on the occurrence of partial remission. 

Healthy controls (Table 2) did not differ from the T1D group by age (*p* = 0.805), and sex distribution (*p* = 0.551). Moreover, siblings and non-siblings among controls were of similar age (*p* = 0.228), did not differ significantly by sex-distribution (*p* = 0.644) or by SAF values (*p* = 0.883). 

### 3.2. Variables at the Time of DM Diagnosis

Already at the time of diagnosis, subjects with T1D had significantly higher SAF compared with controls (Table 3).

The two-factor ANOVA indicated a significant effect of the presence of diabetes (*p* < 0.001), but not that of sex (*p* = 0.320) on SAF. For SAF as a primary end-point, the post-hoc power analysis indicated power of 98.8% at α = 0.05. No sex differences in SAF were revealed either in the T1D group (*p* = 0.163) or in controls (*p* = 0.833). At the time of diagnosis, SAF positively correlated with age both in the T1D group (Y = 0.0195X + 0.8522, r = 0.374, *p* < 0.001) and in controls (Y = 0.0224X + 0.7040, r = 0.518, *p* < 0.001) (Figure 3). 

SAF values did not significantly differ in children with or without ketoacidosis (1.01 ± 0.25 vs. 1.00 ± 0.22 AU, *p* = 0.801). SAF values significantly correlated with HbA1c (r = 0.285, *p* = 0.002) (Figure 4A), but not with C-peptide (r = 0.034, *p* = 0.721) (Figure 4B), and C-peptide/glucose ratio (r = 0.083, *p* = 0.377) (Figure 4C) or with daily dose of insulin at the time of discharge from hospital (r = 0.074, *p* = 0.433) (Figure 4D).

### 3.3. Follow-Up Study

SAF measured at the time of DM diagnosis significantly correlated with delta HbA1c (e.g., the difference between the follow-up and baseline values; r = 0.225, *p* = 0.016) (Figure 5A), but not with HbA1c after one year (r = 0.120, *p* = 0.204) (Figure 5B), or mean HbA1c during the first year (r = 0.181, *p* = 0.054) (Figure 5C). Moreover, SAF correlated with C-peptide (r = 0.260, *p* = 0.005) (Figure 5D) and C-peptide/glucose ratio (r = 0.263, *p* = 0.005) (Figure 5E), but not with total daily insulin dose (r = 0.014, *p* = 0.878) (Figure 5F).

At follow-up, SAF values were similar to those at diagnosis (n = 56; 0.99 ± 0.24 vs. 0.95 ± 0.14 AU, *p* = 0.198).

In the forward linear multiple regression analysis, HbA1c change within the first year of T1D duration was associated only with HbA1c at the time of DM diagnosis (∆R^2^ = 0.786, *p* < 0.001). C-peptide after one year of T1D duration was associated with age (∆R^2^ = 0.413, *p* < 0.001) and HbA1c at DM diagnosis (∆R^2^ = 0.021, *p* = 0.025), and C-peptide/glucose ratio after 1 year of T1D duration also with age (∆R^2^ = 0.401, *p* < 0.001) and HbA1c at DM diagnosis (∆R^2^ = 0.020, *p* = 0.029) (Table 4).

### 3.4. Effects of Diabetic Ketoacidosis

At the admission to the hospital, 53 individuals (46%) presented with DKA. Patient with DKA were younger (6.7 ± 4.4 vs. 9.1 ± 4.3 years, *p* = 0.004), displayed higher HbA1c (12.6 ± 2.1 vs. 11.4 ± 2.5%; 114 ± 23 vs. 101 ± 27 mmol/mol, *p* = 0.006), lower C-peptide levels (110 (72–158) vs. 167 (98–275) pmol/L, *p* = 0.001), while they did not differ significantly by SAF values (1.01 ± 0.25 vs. 1.00 ± 0.22 AU, *p* = 0.801). In both groups HbA1c values declined during the follow-up, they differed significantly between the groups only at the 3rd month (Figure 6). After 12 months, the subject who did not suffer from DKA at diagnosis still had higher C-peptide levels (176 (38–395) vs. 89 (3–230) pmol/L, *p* = 0.022).

### 3.5. Remitters vs. Non-Remitters Group

At the time of diagnosis, remitters presented lower SAF compared with non-remitters (0.95 ± 0.18 vs. 1.04 ± 0.26 AU, *p* = 0.027) (Figure 7), albeit in both cases significantly higher (remitters: *p* = 0.048; non-remitters: *p* < 0.001) compared with controls (0.88 ± 0.19 AU, Table 1).

Remission occurred in 44 patients (38.6%, CI 30.2–47.8). Compared with non-remitters, remitters had higher BMI SDS at T1D diagnosis, higher age, higher C-peptide, higher C-peptide/glucose ratio, and lower insulin daily requirements at the time of discharge from hospital. During the first year, their mean HbA1c was lower, thus the change in HbA1c was higher. The groups did not differ in HbA1c levels at the time of DM diagnosis.

At follow-up, remitters displayed higher C-peptide, C-peptide/glucose ratio, and had lower insulin daily requirements. 

SAF at the time of diagnosis did not differ from that at follow-up, regardless of presence (0.98 ± 0.18 vs. 0.97 ± 0.15 AU, *p* = 0.804) or absence (1.00 ± 0.26 vs. 0.94 ± 0.13 AU, *p* = 0.204) of remission. 

In a forward multiple logistic regression, the occurrence of remission was associated with SAF (∆R^2^ = 0.051, *p* = 0.004) and age (∆R^2^ = 0.089, *p* = 0.007) at the time of diagnosis, but not with HbA1c (Table 5). ROC analysis displayed AUC of 0.625 (95% CI: 0.524–0.729), *p* = 0.024.

## 4. Discussion

The main findings of our study are that: A/individuals with T1D present already at the time of diagnosis significantly higher SAF compared with controls; B/in T1D individuals, SAF at the time of diagnosis associates with several markers of the β-cell secretory capacity; and C/SAF at the time of DM diagnosis is not affected significantly by the presence or absence of ketoacidosis but is an independent predictor of the remission. 

It is well documented that children and adolescents with a mean age of 12-to-16 years and a mean duration of T1D between 4-to-9 years present significantly higher SAF compared with their healthy siblings or age-matched controls [12,14,20,21]. Our data extend this knowledge as that SAF is higher already at the time of T1D diagnosis, even in toddlers and small children. Albeit our patients who later on manifest a remission presented at diagnosis with lower SAF values compared with their non-remitter counterparts, still their levels were higher compared with healthy controls. In the control group, SAF did not differ significantly between siblings and non-siblings. In line with other studies, we also confirmed that SAF increases with age regardless of sex or the presence or absence of T1D [14,21,27]. 

Moreover, as in children and adolescents with a longer duration of T1D [12,14,20,21], SAF values correlated directly with HbA1c also in our cohort of newly diagnosed young children, albeit correlation coefficients were much weaker. This might reflect the fact that in untreated diabetes with a short duration, SAF—a long term marker of hyperglycemia—lags behind potentially marked excursions of HbA1c levels. 

However, SAF at DM diagnosis was not significantly associated with markers of further T1D compensation during the first year, such as mean HbA1c, and HbA1c at 1st year of follow-up. This suggests that regardless of SAF values at the time of diagnosis, adequate insulin treatment, and adherence to the proposed regimen enabling good metabolic control, potentially prevents a further rise of SAF within the first year after the diagnosis. 

Ketoacidosis-induced metabolic stress imposed by hyperglycemia, hyperketonemia, increased production of reactive oxygen species, and α-dicarbonyls [27,28] leads to the acute formation of early glycation products and later production of AGEs. Accumulation of Amadori products is manifested currently with the onset of DKA, as reflected by elevated HbA1c at diagnosis. Despite an intensive insulin treatment, HbA1c levels remained higher up to the 6th month in the DKA group compared with their peers without DKA. The slow decline reflects the half-life of red blood cells on one hand and the fact that glycation of hemoglobin is a faster reaction compared with the reverse one [29]. As the formation of AGEs requires a longer time, it is not surprising that the presence or absence of diabetic ketoacidosis at admission was not mirrored by differences in SAF. Even though the SAF values did not differ between the groups at follow-up, a transitional increase cannot be excluded. 

Dynamics of AGEs accumulation after the recovery from DKA could be elucidated by sequential measurements of SAF, plasma, or salivary AGEs. 

Our study is the first to examine the association of SAF with parameters of the residual secretory capacity of pancreatic B-cells at the time of DM diagnosis, as well as during the first year of follow-up. Residual B-cell secretory capacity depends on several factors, including the early DM diagnosis, the presence of DKA at DM diagnosis, as well as the age of the child, and the presence of autoantibodies against pancreatic B-cells. The fact that several factors not included in our study might modulate the residual pancreatic function, probably underlie the observed weak associations with SAF. HbA1c values have also been associated with these parameters in several studies [30,31]. As SAF reflects an accumulation of AGEs, and the half-life of AGEs is much longer than that of HbA1c, we hypothesized that SAF might be a better predictor of the residual B-cell secretory capacity, as well as remission period, and DM compensation within a year after diagnosis.

Although SAF at the time of diagnosis was significantly associated with C-peptide and C-peptide/glucose ratio values after one year of T1D duration in univariate analyses, it was not an independent predictor in multiple regression models. This reflects the fact that there are several factors affecting residual B-cell function already at the time of diagnosis, which were not assessed in our study.

Unlike HbA1c, SAF values were independent predictors of the development of remission in children and adolescents with DM1. This data suggest that having appropriate clinical data on SAF in pediatric age categories of patients diagnosed with T1D with or without DKA might allow for establishing age-dependent SAF cut-offs to predict the potential of remission. 

The main strengths of our study are the inclusion of newly diagnosed T1D patients covering a wide age range. All of the participants were diagnosed and followed up in a single center. 

The limitation is that Slovak legislation allows for starting insulin pump therapy at the earliest after the 6th month of diabetes duration. Moreover, at the time of our study Slovak Health Insurance companies did not cover a continuous glucose monitoring system. It is to be mentioned that the patients were managed by four pediatric diabetologists (LB, EJ, JS, and KP).

## 5. Conclusions

Our study brings new insight into the potential application of SAF determination in diabetology. While in adult patients SAF is a predictor of chronic micro-and macrovascular complications, in young subjects just diagnosed with diabetes and not suffering from comorbidities yet, it rather seems to be a predictor of remission. Further longitudinal multicenter studies starting at the time of T1D diagnosis covering the whole pediatric age range will be needed to define the clinical potential of SAF measurement.

## Figures and Tables

**Figure 1 ijerph-19-11950-f001:**
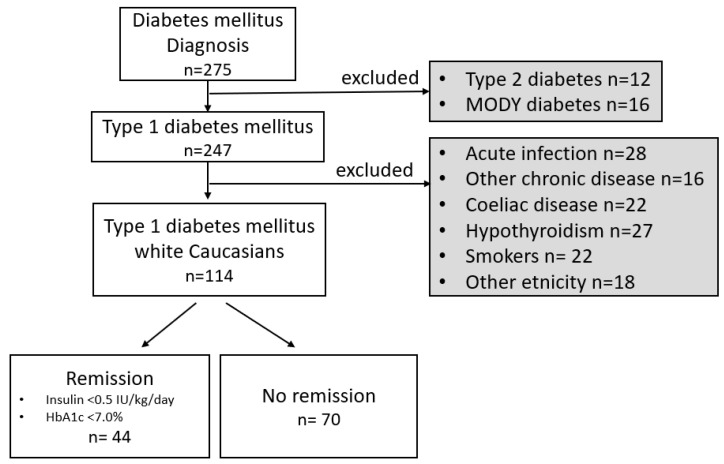
Flow chart documenting inclusion criteria, exclusion criteria, and the number of individuals included in the study population. **Abbreviations**: MODY—Maturity Onset Diabetes of the Young, IU—International Units, HbA1c—glycated hemoglobin A1c, remission was classified according to ISPAD guidelines 2018 as total daily dose of insulin < 0.5 IU/kg/day and HbA1c < 7.0% (53 mmol/mol).

**Figure 2 ijerph-19-11950-f002:**
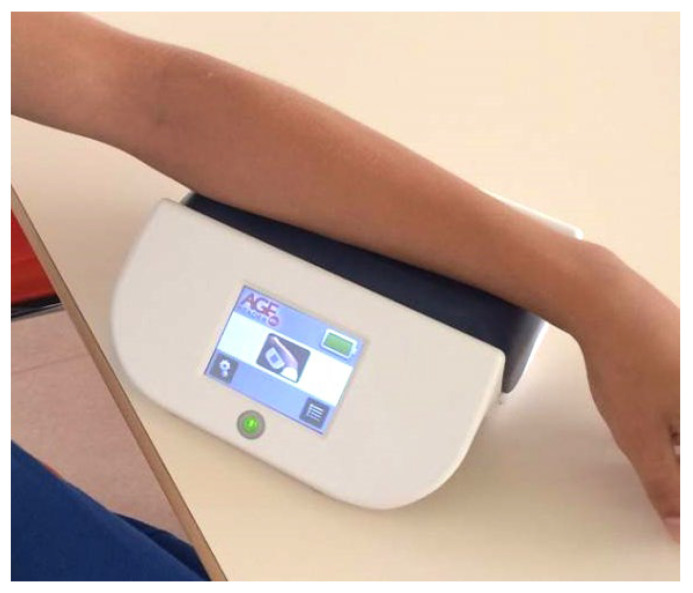
Practical measurement of skin autofluorescence.

**Figure 3 ijerph-19-11950-f003:**
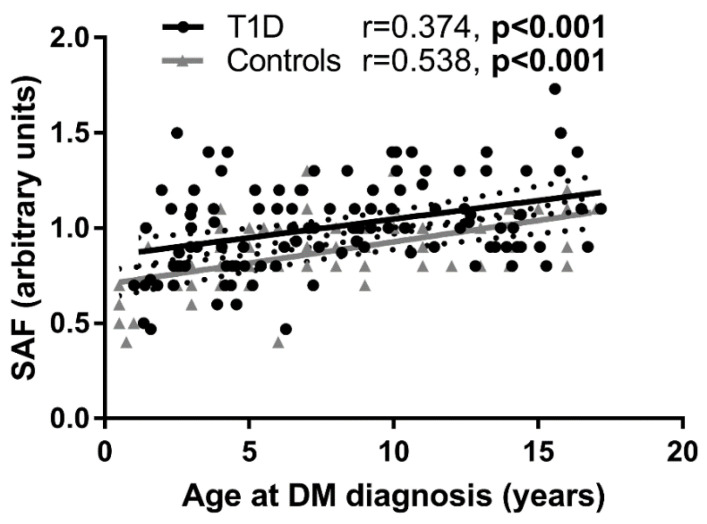
Pearson’s correlation of SAF with age in children and adolescents with newly diagnosed T1D and controls. Dotted lines represent the 95% confidence intervals of the regression line calculated with linear regression analysis. SAF correlated with age at T1D diagnosis in T1D group (Y = 0.01949X + 0.8522, r = 0.374, *p* < 0.001) and with age at examination in controls (Y = 0.02237X + 0.704, r = 0.518, *p* < 0.001).

**Figure 4 ijerph-19-11950-f004:**
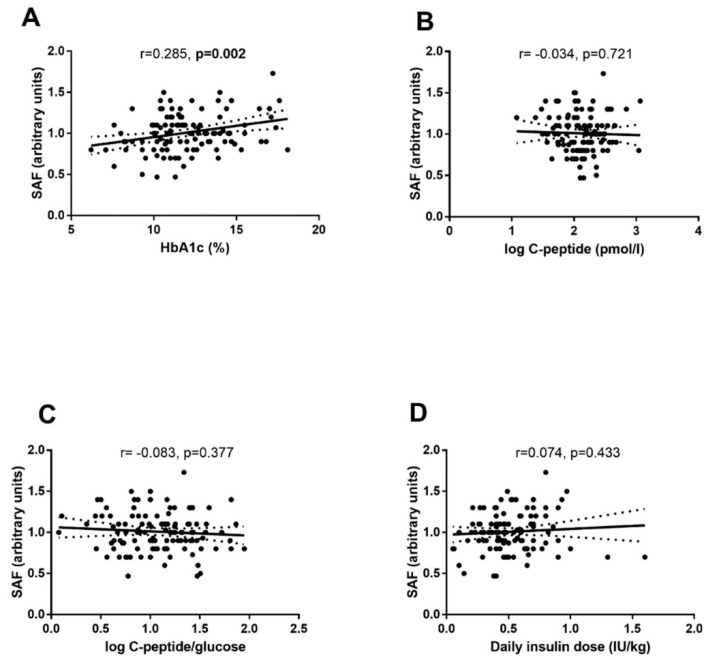
Association of SAF at the time of diagnosis with selected parameters at the time of DM diagnosis. Pearson’s correlations of SAF with HbA1c (**A**), fasting C-peptide serum levels (**B**), fasting C-peptide/glucose ratio (**C**), a daily dose of exogenous insulin at the discharge from hospital (**D**). Dotted lines represent the 95% confidence intervals of the regression line calculated with linear regression analysis.

**Figure 5 ijerph-19-11950-f005:**
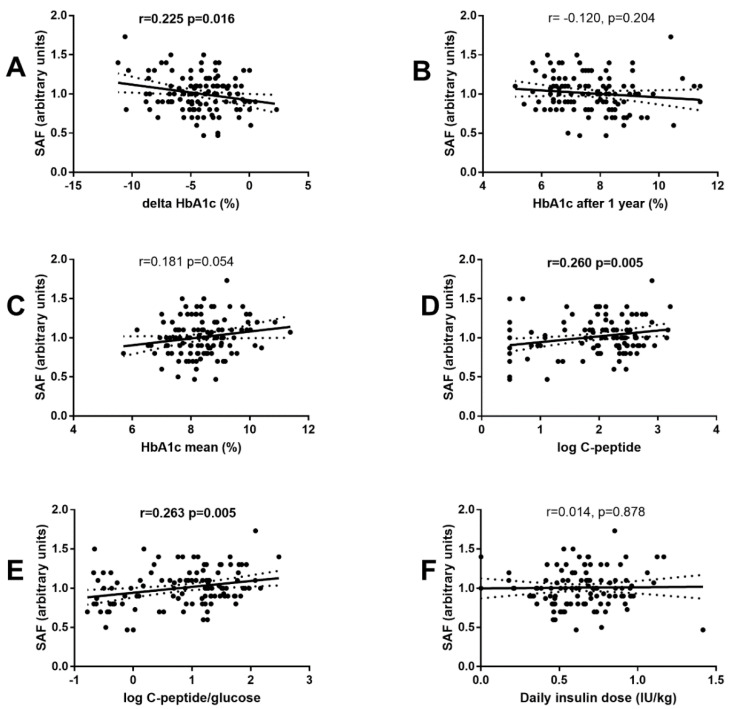
Association of SAF measured at the time of T1D diagnosis with selected parameters during the first year of T1D follow-up. Association of SAF at the time of DM onset with HbA1c change during the first year of T1D duration (**A**), HbA1c after the first year of T1D duration (**B**), mean HbA1c during the first year of T1D duration (**C**), fasting C-peptide serum levels after the first year of T1D duration (**D**), fasting C-peptide/glucose ratio after the first year of T1D duration (**E**), a daily dose of exogenous insulin after the first year of T1D duration (**F**). Pearson’s correlation was used in (**A**–**E**). Dotted lines represent the 95% confidence intervals of the regression line calculated with linear regression analysis.

**Figure 6 ijerph-19-11950-f006:**
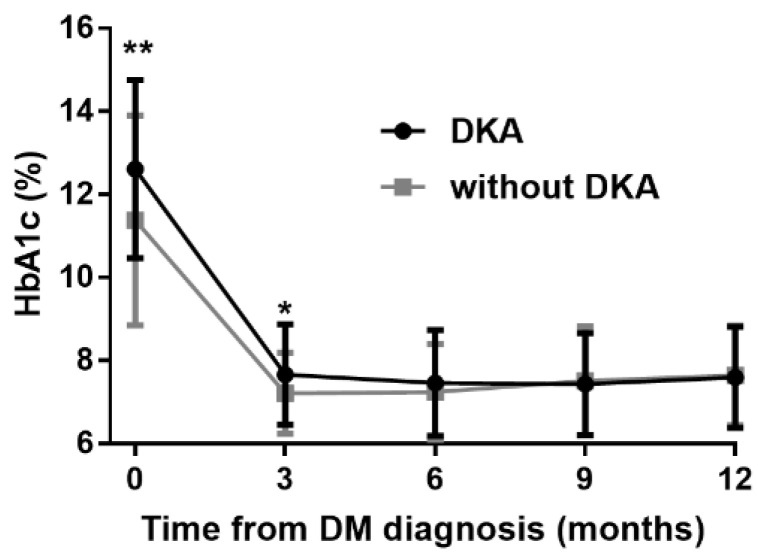
Concentrations of HbA1c during the first-year follow-up in patients who. manifested or did not manifest with DKA, *: *p* < 0.05, **: *p* < 0.01. Differences were calculated using the *t*-test. Data are expressed as mean ± SD.

**Figure 7 ijerph-19-11950-f007:**
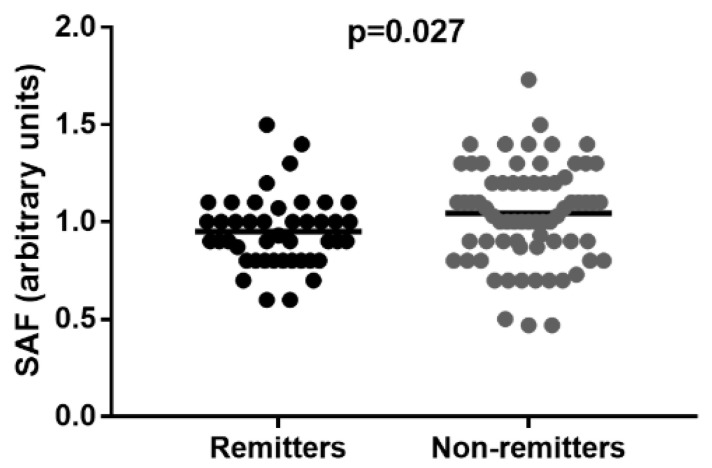
Comparison of SAF values in remitters and non-remitters, lines represent mean. Differences were calculated using the *t*-test.

**Table 1 ijerph-19-11950-t001:** Basic characterization of children and adolescents with newly diagnosed T1D and comparison between remitters and non-remitters.

	All	No Remission	Remission	*p*
n (%, CI)	114	70 (61.4, CI: 52.2–69.8)	44 (38.6, CI: 30.2–47.8)	NA
Sex, boys (%)	64 (56)	39 (56)	25 (57)	1.000
AT DIABETES DIAGNOSIS				
BMI SDS	−0.5 ± 1.1	−0.7 ± 0.9	−0.1 ± 1.0	<0.001
Age, years	8.0 ± 4.5	7.3 ± 4.3	9.1 ± 4.7	0.041
HbA1c, % (mmol/mol)	11.9 ± 2.4 (107 ± 27)	12.0 ± 2.6 (108 ± 28)	11.9 ± 2.2 (106 ± 24)	0.786
C-peptide, pmol/L	133.5; 80.5–219.5	121.0; 65.2–193.7	158.5; 99.7–255.7	0.019
C-peptide/glucose ratio	13.7; 6.5–24.8	10.3; 5.5–22.1	14.8; 9.8–27.0	0.032
Diabetic ketoacidosis, yes (%)	53 (46)	35 (50)	18 (41)	0.441
Antibodies, positive (%)	107 (94)	66 (94)	41 (93)	1.000
Skin autofluorescence, arbitrary units	1.01 ± 0.23	1.04 ± 0.26	0.95 ± 0.18	0.027
Insulin daily dose at discharge (IU/kg/24 h)	0.5 ± 0.2	0.6 ± 0.3	0.5 ± 0.2	0.034
Duration of remission (days)	NA	NA	201 ± 199	0.001
FOLLOW-UP AFTER 1 YEAR OF DIABETES DURATION				
HbA1c, % (mmol/mol)	7.6 ± 1.2 (60 ± 13)	7.9 ± 1.2 (63 ± 13)	7.1 ± 1.0 (54 ± 11)	0.236
Delta HbA1c, % (mmol/mol)	−4.3 ± 2.6 (−47 ± 29)	−4.1 ± 2.8 (−45 ± 31)	−4.7 ± 2.2 (−50 ± 25)	<0.001
Mean HbA1c, % (mmol/mol)	8.4 ± 0.9 (68 ± 11)	8.7 ± 0.9 (72 ± 10)	7.8 ± 0.8 (62 ± 9)	<0.001
C-peptide, pmol/L	121.0; 12.7–297.0	78.0; 9.2–202.5	263.5; 99.2–425.7	<0.001
C-peptide/glucose ratio	13.7; 1.4–30.4	6.5; 1.0–17.6	20.4; 9.7–53.8	<0.001
Insulin daily dose, IU/kg/24 h	0.6 ± 0.2	0.7 ± 0.2	0.5 ± 0.2	<0.001

BMI—body mass index; SDS—standard deviation score, NA—not applicable, OR—odds ratio, CI—confidence intervals; delta HbA1c—a difference in HbA1c values at the time of T1D diagnosis and at the follow-up visit after 1 year; mean HbA1c—an average concentration calculated using all obtained values during the first year of the follow-up; insulin daily dose—dose applied at 1st year follow-up visit. Data are expressed as mean ± SD (normally distributed data) or as median and interquartile range. Differences between the two groups were tested using two-sided Student’s *t*-test for normally distributed and Mann–Whitney U test for non-normally distributed metric data, and by Fisher’s test for binary data.

**Table 2 ijerph-19-11950-t002:** Baseline characterization of controls.

	All (n = 74)	Siblings (n = 43)	Non-Siblings (n = 31)	*p*
**Age, years**	7.9 ± 4.5	7.3 ± 3.9	8.6 ± 5.1	0.228
**Sex, boys (%)**	38 (51)	21 (49)	17 (55)	0.644
**SAF (arbitrary units)**	0.88 ± 0.19	0.88 ± 0.2	0.88 ± 0.18	0.883

Values of measurable variables are given as mean ± SD. Differences between the two groups were tested using two-sided Student’s *t*-test for metric data, and by Fisher’s test for binary data.

**Table 3 ijerph-19-11950-t003:** Skin autofluorescence at the time of diagnosis in both groups.

	T1D (n = 114)	Controls (n = 74)	*p*
**Age, years**	8.0 ± 4.5	7.9 ± 4.5	0.805
**Sex, boys (%)**	64 (56)	38 (51)	0.551
**SAF (arbitrary units)**	1.01 ± 0.23	0.88 ± 0.19	*p* < 0.001

T1D—Type 1 diabetes mellitus, SAF—skin sutofluorescence. Differences between the two groups were tested using two-sided Student’s *t*-test for metric data, and by Fisher’s test for binary data.

**Table 4 ijerph-19-11950-t004:** Multiple forward linear regression of selected T1D parameters as dependent variables.

Step	Parameter	ΔR^2^	Standardized β	*p* Value
Independent variables: age, HbA1c, and SAF at DM diagnosis				
**Dependent variable: HbA1c change in 1 year of DM duration** **(R^2^ = 0.786; *p* < 0.001; n = 113)**				
**1**	HbA1c	0.786	0.232	<0.001
Dependent variable: C-peptide after 1 year of DM duration (R^2^ = 0.434; *p* < 0.001; n = 114)				
**1**	age	0.413	0.601	<0.001
	HbA1c	0.021	0.167	0.025
**Dependent variable: C-peptide/glucose ratio after 1 year of DM duration** **(R^2^ = 0.421; *p* < 0.001; n = 114)**				
**1**	age	0.401	0.593	<0.001
**2**	HbA1c	0.020	0.165	0.029

**Abbreviations**: CI—95% confidence intervals. SAF—skin autofluorescence.

**Table 5 ijerph-19-11950-t005:** Logistic regression of remission as dependent variable.

Step	Parameter	ΔR^2^	β (CI)	*p* Value
**Dependent variable: remission (R^2^ = 0.140; *p* = 0.002; n = 114)**				
Independent variables: age, HbA1c, and SAF at DM diagnosis				
**1**	SAF	0.051	0.051 (0.007–0.390)	0.004
**2**	age	0.089	1.151 (1.040–1.273)	0.007

CI—95% confidence interval; SAF—skin autofluorescence; DM—diabetes mellitus

## Data Availability

The data presented in this study are available on request from the corresponding author.

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
