# Peer review of "The Bright Side of Skin Autofluorescence Determination in Children and Adolescents with Newly Diagnosed Type 1 Diabetes Mellitus: A Potential Predictor of Remission?"

_ijerph, 2022, doi:10.3390/ijerph191911950_

Round 1

Reviewer 1 Report

In this article, Podolakov et al. identified Skin autofluorescence (SAF) as an independent prognostic marker in newly diagnosed type 1 diabetes mellitus in children and adolescents. The analysis was carefully performed some clarifications are needed to strengthen the rationale of the study and the conclusions. Comments are as follows:

Comments:

1. In the introduction part, the authors did not describe much about type 1 diabetes mellitus. Since it is the main theme of the paper, more information regarding this will be helpful.

2. A flow chart for the inclusion and exclusion criteria for the study design will be helpful and easy to follow.

3. In the follow-up study the authors used delta Hb1Ac (section 3.3) although in the primary results they did not show results for delta Hb1Ac. What is the status of delta Hb1Ac before (Figure 2) and what’s its importance as compared to Hb1Ac alone?

4. In section 3.5 the authors compared SAF between remitters and non-remitters. It seems the remitters still have higher SAF values as compared to healthy controls. Can the authors compare remitters vs. the control group and comment on the outcomes?

Author Response

First of all we would like to thank the reviewer for his time and review. 

Podoláková

Reviewer 2 Report

This paper focuses on the study of the association between skin autofluorescence and  T1D patients at the time of DM diagnosis and first year of follow up together with other markers. The results show a very interesting information which can be useful for the people in the field. However, there are few points that need further clarification;

1. The study has a sample size of 114 children and adolescents. The calculation of the sample size should be shown to verify that the sample size is big enough for the study.

2. As the skin autofluorescence (SAF) was used for this study, a paragraph to describe how the SAF works is encouraged. The image showing the SAF measurement on the volar side of the dominant forearm should be provided. For example, In figure 5, when a 'lower SAF' was mentioned, does it mean an intensity of the fluoresence? It's not clear. The paragraph to explain the SAF and how the data can be recored would help for a better understanding.

3. Please explain 'For technical reason, at one year SAF was measured only in 56 patients with T1D. What does it mean by technical reason?

4. Is it possible to put the table similarly to tables 1 and 2  for the sections 3.2 and 3.3 as it will beeasier to follow up

5. For figures 2 and 3, some of the results show relatively low correlation values e.g. 0.285, 0.225, 0.260 and 0.263 indicating low correlation between two parameters. However, they all have p-value less than 0.05 which is considered to be significant. The authors conclude that the two parameters have significantly correlation. It is not convincing as the R-value is too low. Please give more explaination on this matter.

Author Response

(The authors gave the same response as above.)
